# Assessing the Efficacy of a Brief Universal Family Skills Programme on Violence and Substance-Use Indicators in Youth in Trentino and Parma, Italy: Study Protocol for a Multi-Centre, Non-Blinded, Cluster-Randomised Controlled Trial (cRCT) of Family UNited

**DOI:** 10.3390/ijerph20166548

**Published:** 2023-08-08

**Authors:** Karin Haar, Aala El-Khani, Riccardo Lodi, Valentina Molin, Annalisa Pelosi, Ali Yassine, Giovanna Campello, Wadih Maalouf

**Affiliations:** 1Prevention, Treatment and Rehabilitation Section (PTRS), Drugs, Laboratory and Scientific Services Branch (DLSSB), Division for Policy Analysis and Public Affairs (DPA), United Nations Office on Drugs and Crime (UNODC), Wagramer Strasse 5, A-1400 Vienna, Austria; karin.haar@un.org (K.H.); aala.el-khani@un.org (A.E.-K.); ali.yassine@un.org (A.Y.); giovanna.campello@un.org (G.C.); 2S.O.L.E. Italia, Strada Corte Delle Grazie 21, 43126 Parma, Italy; riccarlodi@gmail.com; 3Department of Sociology and Social Research, University of Trento, Via Tarter 2 Baselga di Piné, 30122 Trento, Italy; valentinamolin@yahoo.it; 4Neurosciences Unit, Department of Medicine and Surgery, University of Parma, Via Volturno 139, 43125 Parma, Italy; annalisa.pelosi@unipr.it

**Keywords:** cluster-randomised controlled trial (cRCT), study protocol, family skills programme, parenting skills, violence and substance use in youth, Italy

## Abstract

Homes in which families are experiencing stressful and challenging circumstances can foster a social space that engenders violent behaviours in parents, inadequate childcare, and the exposure of children to criminal and antisocial behaviours at an early age in addition to many other negative social and health consequences throughout their development. Family Skills Training offers a combination of parenting knowledge, skill building, competency enhancement, and support to strengthen family protective factors, such as communication, trust, problem-solving skills, and conflict resolution. Through over a decade-long experience piloting evidence-based family skills packages globally, we developed a universal open-source family skills package, “Family UNited” (FU), designed for families with children aged 8 to 15 years living in low- and middle-income countries (LMIC). The current study aims to explore the efficacy, fidelity, and acceptability of FU in Trentino and Parma, Italy. We plan to conduct a multi-site, non-blinded, two-armed, cluster-randomised controlled trial to assess efficacy in 160 families: the intervention group receiving FU and the waitlist/control group only receiving FU after the completion of all data collection points. We will prospectively collect outcome data, assessing changes in parenting skills and family adjustment in caregivers, children’s behaviour, resilience capacities, and attitudes towards peer violence. To assess programme delivery, fidelity, feasibility, and acceptability we will include an embedded process evaluation. This study aims to evaluate the improvement in parenting skills, child well-being, and family mental health after participation in FU, compared to no intervention. Even though this trial is to be conducted in a high-income country, such results complement the existing piloting experience in LMIC. with impact-related measures encouraging the adoption of such approaches globally and beyond the EU borders.

## 1. Introduction

### 1.1. Background and Rationale

Family Skills Training offers a combination of parenting knowledge, skill building, competency enhancement, and support [1]. Family skills packages have been designed to reduce current family relationship problems and prevent future difficulties through training, support, or education [2]. They aim to strengthen family protective factors, such as communication, trust, problem-solving skills, and conflict resolution, and often include opportunities for parents and children to spend positive time together as a way to strengthen the bonding and attachment between the two.

Family skills training programmes have frequently been used at a universal level of prevention. As such, they target whole populations, most frequently through schools, neighbourhoods, or communities, without specific screening at the individual level for the level of risk presented. Such universal approaches share the assumption that almost anyone can benefit from such a prevention and health promotion/orientation package, and that delivery to groups made up of families with differing levels of risk is likely to foster engagement. Such an approach also has the potential of reducing any potential stigmatization of the programme, in the sense of associating it with targeting families with specific problems and facilitating as many families as possible to benefit from it. Nevertheless, such a universal approach does not necessarily mean that the package affects all families equivalently; but, the aim is always to mostly benefit those families facing higher levels of challenges. Individual cases with higher levels of problems will be identified by facilitators and referred to more tailored or individualised services [3].

### 1.2. The Role of the Family in Violence and Substance-Use Prevention

Homes in which families are experiencing stressful and challenging circumstances can foster a social space that engenders violent behaviours in parents, inadequate childcare, and the exposure of children to criminal and anti-social behaviours at an early age in addition to many other negative social and health consequences throughout their development. When parents are harsh in discipline and low in responsiveness, this accelerates the negative behaviours of the youth and serves as an impetus for criminal behaviours and gang involvement; whereas, warmth and positive parenting can prevent negative social outcomes, such as the initiation of drug use and other risky behaviours [4,5].

### 1.3. The Family UNited Programme

Through its over a decade-long experience piloting evidence-based family skills packages globally, the United Nations Office on Drugs and Crime (UNODC) has identified that very few family skills programmes were designed to serve the needs of families living in low-resource settings (the primary targets settings of UNODC) [6]. Nevertheless, the few available current reports appear promising and indicate the feasibility of this approach in such settings [7]. Based on this knowledge, the UNODC Prevention Treatment and Rehabilitation Section (PTRS) initiated open-sourced family skills packages designed for low- and middle-income countries (LMIC) to protect youth from vulnerabilities. The UNODC’s first experience was with the Strong Families programme, which was targeted to families living in challenged and humanitarian settings. The feasibility and impact of Strong Families have been demonstrated in families living in three cities in Afghanistan [8], in families residing in refugee reception centres in Serbia [9], and, most recently, through a multi-site randomised controlled trial in Iran [10]. It is currently being implemented in a number of other countries globally.

With the evolving experiences with Strong Families, the Families UNited programme began taking shape. Family UNited draws from three overarching theories which shaped the components of the programme sessions. Firstly, the Biopsychosocial Vulnerability Model [11] suggests that positive family coping skills, such as conflict resolution, active problem-solving skills, and positive communication, shield individual family members and protect youth from vulnerabilities stemming from the negative effects of family conflicts. In this theory, caregivers’ influence on their children is greatest when the children are younger and decreases significantly as they enter early adolescence. The second theory is the Resiliency Model [12], which emphasises the foundational role caregivers in a family play in children developing resilience. Resilience is defined as the ability to rebound from difficult or adverse circumstances [13] and is thought to be more likely to develop in children when raised in a family environment in which caregivers are both positive and supportive [14]. This theory focuses on life skills that are promoted when caregivers are supportive, such as reflective skills, emotional management skills, and the ability to problem solve. This theory is supported by research that identifies that the relationship a child has with their caregiver can have a more significant impact on their mental health trajectory than the experiences of war and displacement [15]. The third theory is the Social Learning Theory [16], which proposes that children’s daily experiences of the world through their interactions with others, imitation, and the reinforcement they receive, shape their behaviour both directly and indirectly [17]. This places the role of caregivers as pivotal for their healthy social development and also guides family skills interventions to focus on improving the quality of parenting by improving foundational parenting skills [18].

Based on its theoretical foundations, the content of the Family UNited Programme focuses on enhancing an empathetic and warm approach to caregiving; improving family cohesion, communication, and relationships; and gaining skills for emotional regulation and assertive skills for managing peer pressure. There is no content addressing substance use explicitly.

Family UNited is a universal prevention intervention designed for families with children aged between 8 and 15 years with the aim of improving parenting skills, child well-being, and family mental health.

A logic model based on this approach is shown in Figure 1 [19].

Overall, the Family UNited Programme consists of four sessions (one per week) addressing the essential core needs of parenting skills. Each session does not exceed two hours, with the first hour addressing caregivers and children separately and the second hour working with the whole families (Table 1).

So far, Family UNited has been piloted in Bangladesh and Indonesia, with promising results regarding child mental health, resilience, and parenting skills (data have been submitted for peer-reviewed publication). In addition, facilitators have been trained (remotely) in Karakalpakstan (Uzbekistan); and, an online training platform for facilitator training is being developed. Further piloting studies with families have so far had to be postponed due to COVID-19 social distancing measures in many countries.

Being a universal, evidence-informed family skills package with an overall design vision that is deemed easy to implement in low-resource settings also gives Family UNited the advantage of the ease of scalability in low- as well as higher-resource settings. It is in that context that researchers from the University of Parma and Trento were interested in piloting this package in their respective locals. To our knowledge, no other family skills programmes have been implemented in Parma or Trentino before.

### 1.4. Objectives of This Study

The current study aims to explore the efficacy, fidelity, and acceptability of this universal prevention programme with families in Trentino and Parma, Italy and investigate the feasibility of delivering the programme.

The primary objective (impact evaluation) will be to test the efficacy of the Family UNited programme in improving family-skills outcomes and caregiver and child mental health and resilience, as reported by caregivers and children, when implemented in Trentino and Parma, Italy. According to the logic model of Family UNited (Figure 1), the short-term participant and family impact will be measured and compared against a waitlist/control group.

The secondary objective (process evaluation) will be to calculate the extent of families’ attendance to Family UNited sessions to evaluate the completeness of programme delivery. Further, the fidelity will be assessed to ensure facilitator compliance with the programme, as trained.

The tertiary objective will be to explore the cultural and contextual acceptability of the Family UNited programme for families in Trentino and Parma, Italy.

### 1.5. Trial Design

To explore feasibility and ensure adequate implementation modalities, a pilot study was conducted in March/April 2022.

Subsequently, we plan to conduct a multi-site, non-blinded, 1:1 cluster-randomised controlled trial (cRCT) with two arms used to assess efficacy: the (1) intervention group (receiving the Family United programme) and (2) waitlist/control group (families only receiving the Family UNited programme after the completion of all data collection points).

We will prospectively collect all outcome data, assessing changes in parenting skills and family adjustment in caregivers, children’s behaviour, resilience capacities, and attitudes towards peer violence.

To assess programme delivery, fidelity, feasibility, and acceptability, we aim to include an embedded process evaluation.

## 2. Materials and Methods

### 2.1. Study Setting

This study will take place in different schools in Parma city and Trento province, Italy. The exact location will be decided by the Italian researchers of this study after conducting local assessments for the suitability of the location. Parma city has a population of 194,417; whereas, 538,223 persons live in the Trento province. Parma and Trento are two rich cities in the Italian national scene, located in two of the wealthiest and most-developed regions of Italy, Emilia-Romagna and Trentino-Aldo Adige, with a GDP per capita of 33.6 and 36.9 (in thousand EUR), respectively [20]. In both cities, there are problems of youth addiction due to a number of possible reasons, including the social disintegration of a large number of immigrants, including second-generation immigrants [21]. In their study, Deolmi et al. indicate that 31% of the participants with mental disorders, which includes drug use and addiction, are of immigrant origins [21]. This could be partly explained by the difficulties in integration faced by immigrants when settling down in host countries [22].

### 2.2. Eligibility Criteria

Participants will be parents or caregivers and one child under their care aged between 8 and 15 years. Sampling will be opportunistic, using a ‘universal’ approach in which facilitators will recruit families from the general population, not targeting specifically those with particular difficulties or risks. The inclusion criteria are caregivers who speak Italian and have at least one child under their care, aged 8 to 15 years, willing to take part in the programme and those in the area for the duration of the whole study. Families who took part in any other family skills training programme in the past 24 months, or where the caregiver lives separately from the child or does not have a constant relationship of attendance with the child (i.e., less than half of the time in cases of shared custody), will be excluded from this study.

### 2.3. Intervention Condition

Family UNited will be delivered to families in groups of approximately 12 families. Only one or two parents or main caregivers will be invited to attend, with a maximum of two children under their care aged 8 to 15 years.

The programme consists of four meetings (Table 1) spread over four weeks. In the first hour of each week, two facilitators will work with caregivers in one room; whereas, the other two facilitators will work with the children in a separate room. In the second hour of the meeting, all four facilitators and all caregivers and children will come together in one room for the joint family session.

During the caregiver sessions, caregivers will learn about children’s behaviour and how caregiver attention changes children’s behaviour. They will learn techniques to make the behaviour they want to see more likely to happen. They will practice how to communicate with their children in a way that is healthy and promotes listening. They will also be shown ways to make sure their children know that they are loved and, at the same time, how to set limits that help the children stay safe and feel secure.

During the child sessions, children will learn about some good ways to deal with stress and the importance of talking to trusted adults about their feelings. They will think about their caregiver’s responsibilities and have discussions about the positive qualities they want to develop and use as adults. In addition, children will learn and practice ways to deal with peer pressure situations, increasing the likelihood that they will stay healthy and safe.

In the family sessions, caregivers and their children will come together to practice positive and respectful communication. Together, they will play games and participate in activities that encourage positive relationships and build trust. The children will enjoy one-to-one time with their caregivers while talking about the strengths and qualities they share as families and the things they can do together to demonstrate their family values.

Based on the process evaluation, we will monitor participants’ attendance at each of the sessions and calculate the “Reach” and “Dose received”, as shown in Table 2.

During the trial period, participants will be requested not to take part in any other family skills programme, nor will any other contacts be made with the families, apart from the scheduled sessions of the Family UNited programme.

Facilitators in both Trentino and Parma have been trained on Family UNited through 3 full-day in-person workshop in December 2021. They reached pilot families in March/April 2022 and implemented the programme. The same facilitators will implement Family UNited during the planned cRCT.

### 2.4. Waitlist/Control Condition

Participants allocated to the waitlist/control group will only receive the questionnaires at the same time as the families in the intervention group. They will not receive any family skills intervention, nor will any contact be made with facilitators before all data have been collected. Only then the waitlist/control families will start the Family UNited training programme, as shown in Figure 2.

### 2.5. Outcomes

The primary outcome of this trial is to assess the efficacy of Family UNited in improving family skills outcomes and family mental health by assessing caregivers’ parenting skills, parental confidence, and mental health, as well as their children’s behavioural and mental health outcomes, resilience capacities, and aggression and hostile behaviour as short-term impacts, as described in the logic model (Figure 1). To assess this, caregivers and children will complete a battery of questionnaires, right before the intervention (Time 1), two weeks post-intervention (Time 2), and six weeks post-intervention (Time 3); mean/median scores will be compared, as described in the data-collection and proposed-analysis sections.

In addition, semi-structured interviews will be conducted with a select group of caregivers that have completed the Family UNited programme and facilitators who will include experiences and recommendations through a qualitative perspective.

To calculate families’ rates of recruitment and attendance to the programme and to evaluate the fidelity and completeness of the programme delivery, we adapted the framework developed by Linnan and Steckler [23]. This framework includes seven key dimensions that need to be evaluated: context (environmental aspects of the intervention setting), reach (the proportion of participants who received the intervention), fidelity (whether the intervention is delivered as planned), dose delivered and received (the amount of intervention delivered and the extent to which participants responded to it), implementation, (a composite score of reach, dose, and fidelity), and recruitment (methods used to attract participants).

The framework has been previously used to evaluate the SFP 10–14 programme in the UK [24], as well as the Strong Families programme in Afghanistan [8] and Iran [10]. A template of the indicators is shown in Table 2. Observers (independent research assistants) will attend all Family UNited programme sessions, track family attendance for every session, and provide external evaluation feedback based on standardised questionnaires, which have been used previously [8,10]. Facilitators will likewise complete internal evaluation forms, out of which the indicators, as shown in Table 2, will be calculated. To identify key factors influencing attendance and adherence, we will conduct interviews with caregivers, facilitators, researchers, and coordinators; whereas, our tertiary objective, exploring the cultural acceptability of the programme by families, will be assessed through interviews with caregivers.

### 2.6. Participant Timeline

Conducting a pilot study before embarking on the RCT ensured that research assistants were confident with the use of the tools with families and that facilitators had the chance to deliver the programme to families (Table 3). Further, in the pilot study, potential difficulties in delivering the training were noted and the modality of programme delivery was optimised (i.e., recruitment of families, timing of the sessions, room, equipment, etc.).

### 2.7. Sample Size

In the pilot study, 26 caregivers (17 in Parma and 9 in Trentino) completed the data collection before the intervention; whereas, only 18 caregivers (11 in Parma and 7 in Trentino) completed the data collection 2 weeks after the intervention. Overall, 19 children (11 in Parma and 8 in Trentino) completed both data collections. Based on the completion of both time points and excluding missing data, the following sample sizes were estimated. To keep the power at 80%, and to be able to show results for each of the scales, the sample size was calculated to be 40 participants in each group, resulting in a total number of families of 160, as shown in Table 4 and Figure 2.

### 2.8. Recruitment of Families

Families in the Trento province will be invited to the project through advertising in youth gathering centres and centres for families, schools, etc. The recruitment will use both formal channels (e.g., Agency for Social Cohesion of the Autonomous Province of Trento and Schools) and informal channels (e.g., social networks, WhatsApp parent groups, etc.). In Parma, families will be recruited through schools and probably through some companies willing to advertise. The City of Parma and the local health authority will put notices on their respective sites.

### 2.9. Allocation

Once families agree to participate in this study and the first data collection has been completed, they will be allocated to either take the programme immediately (intervention group) or to come back for the next two data collections and, only then, for the programme (waitlist/control group), depending on the school they will attend the programme in. To facilitate the process and to reduce interference between the two groups, the schools will actually be randomised into intervention and control groups using online software (www.sealedenvelope.com; accessed on 28 January 2022). A cluster randomization, instead of a complete randomization, will be used, due to the peculiar logistical needs deriving from the structure of the territory. We are aware that randomization before recruiting participants can influence recruitment and dropout in the control arm. To minimise these issues, we will instruct staff not to reveal family allocation until families have agreed to take part, signed the informed consent form, and completed the baseline data collection. This is an unblinded trial. Research assistants, staff, and families will be aware of participants’ allocated conditions after the first measurements are completed and during the remaining time of the trial.

### 2.10. Data Collection Methods

All participating caregivers and children will complete a battery of measures at three points; one week pre-intervention (t1), two weeks post-intervention (t2), and six weeks post-intervention (t3), either electronically through Google Forms or conventionally through paper-based forms, as shown in Figure 3. At baseline, the Family Demographics Questionnaire (FDQ) will provide information on the age and gender of the caregivers, their marital, educational and work status, and the number of children they care for under the age of 16 years. In addition, caregivers will provide information on the age and gender of their children and their relationship to them. They will further provide information on their country of origin (if not Italy) and how long they have been living in Italy.

The Strengths and Difficulties Questionnaire (SDQ) is a 25-item behavioural screening questionnaire through which we intend to measure potential changes in children related to short-term impacts, such as “Improved child behaviour”, “Reduced aggressive and hostile behaviour”, and “Improved mental health in children”, as outlined in the logical framework (Figure 1). The 25 questions are summed to five subscales, ranging from 0 to 10 points (“Emotional problems scale”, “Conduct problems scale”, “Hyperactivity scale”, “Peer problems scale”, and “Prosocial scale” [Range: 10–0 points, with lower scores indicating lower prosocial skills]) and the “Total difficulties score”, ranging from 0 to 40 points, with higher scores indicating more problems. The SDQ showed good internal consistency of the Total Difficulties Score in 1394 Italian children aged 8–10 years [26].

Through the 30-item Parenting and Family Adjustment Scales (PAFAS) we intend to measure potential changes in caregivers related to short-term impacts, such as “Improved caregiver confidence in family management skills”, “Improved caregiving in parenting skills”, and “Increased capacity to cope with stress”, as outlined in the logical framework in Figure 1. The 30 questions are summed into 7 subscales, 4 parenting subscales (“Parental Consistency” [Range: 0–15], “Coercive parenting” [0–15], “Positive encouragement” [0–9], and “Parent-child relationship” [0–15]) and 3 family adjustment subscales (“Parental adjustment” [Range: 0–15], “Family relationships” [0–12], and “Parental teamwork” [0–9]). Higher scores on each of the scales mean worse outcomes; so, a decrease in scores after the intervention would mean a beneficial effect. The PAFAS subscales have shown good internal consistencies in two Australian samples (ranging from 0.70 to 0.87) and satisfactory construct and predictive validity [27].

Children are requested to fill in the 17-item revised Child and Youth Resilience Measure (CYRM-R), through which we aim to measure the short-term impacts, such as “Reduced aggressive and hostile behaviors”, “Increased capacity to cope with stress”, and “Improved mental health outcomes in children”. The CYRM-R comprises 2 subscales, the “Personal resilience subscale”, ranging from 10 to 50 points, and the “Caregiver resilience subscale”, ranging from 7 to 35 points. The overall scale is then summed out of the two subscales and ranges from 17 to 85 points, with higher scores indicating more resilience. The CYRM-R showed good overall fit statistics and a Rasch analysis indicated good psychometric properties for both subscales [28]. In our trial, we will use the 5-point version for youth aged 10–23 years with simplified language.

Children will further fill in the 14-item “Attitude Toward Interpersonal Peer Violence” (ATIPV) measure, with which we aim to measure “Reduced aggressive and hostile behaviors”, “Increased capacity to cope with stress”, and “Improved mental health outcomes in children”. A higher total score (summarised from each question ranging from 1 to 4 points) means a higher level of knowledge and skills in resolving conflicts non-violently. Through this measure, youth are asked to indicate their opinions or feelings about fighting (defined as physical fights, with pushing and hitting), not just arguments [29].

At t2 and t3, caregivers and children are requested to fill the SDQ, PAFAS, CYRM-R, and ATIPV again, making sure that the same caregiver and child complete the questionnaires at all time points (Figure 3).

All questionnaires will be merged through unique identifiers for each family.

Ten caregivers from each location will be invited to take part in interviews after completion of the Family UNited programme. The criteria for participation will be a randomised sampling of families. A semi-structured approach will be used to reduce the risk of missing any unanticipated issues and schedules; interview guides will provide a checklist of key areas of investigation, with prompts for guiding the interview. Topic areas will explore cultural acceptability, the efficacy of the programme against aims, barriers and facilitators to the engagement of the programme, key factors to adherence, and experience with recruitment.

For the process evaluation, all facilitators of the Family UNited programme will complete reflection sheets on recruitment, dose delivered, reach, staffing, and group size, following each programme session. Independent observers will complete checklists on the implementation of each of the sessions.

All English versions of the caregiver, child, facilitator, and observer questionnaires can be found in the Appendix A; however, in the field, translations into Italian will be used. Previously-used translations have been used for the SDQ [30]; all other questionnaires were translated by bilingual Italian/English researchers for this trial.

### 2.11. Data Management

#### 2.11.1. Data Entry and Data Coding

All non-digital data will be entered electronically using R statistical software or EpiData and electronic data from Google Forms will be added. Analyses will be conducted using commercial statistical programmes, such as Excel, SPSS, STATA, R, etc. For identification purposes, names of the caregivers and children will be collected and will only be used by the research assistants during the intervention and data-collection phase. No identifying data will be entered or stored electronically. All data will be maintained in accordance with the Data Protection Act (1998).

The project managers in Italy, Riccardo Lodi, Annalisa Pelosi, and Valentina Molin, will be responsible for the storage of all digital and non-digital data. Interviews will be audio recorded (with participants’ and facilitators’ permission) using a digital voice recorder supporting file encryption. Notes will be written up as soon as possible following observation sessions and stored with transcriptions of the digital recordings. Identifying material will be removed as soon as possible from the transcripts and notes. Qualitative data will be copied into software files. Electronic copies of the transcripts will be held separately from the digital recordings of interviews and the file containing the participants’ names and corresponding numbers. Qualitative interviews will be conducted and transcribed in Italy; then, translated transcripts will be sent for analysis as encrypted files via electronic mail.

All non-digital data, such as sheets with participant identifiers, notes of the facilitators during the meetings, etc., will be immediately destroyed after the end of this study. All digital data (questionnaires and digitised interviews) will be kept for a maximum of 5 years at the University of Trento.

#### 2.11.2. Data Quality

Digital data entry will avoid entry errors caused by the researcher; although, it will not be able to eliminate the respondents’ compilation errors. Before the analyses, any out-of-scale values and the presence of missing data will be checked. A percentage of missing data greater than 30% will lead to the exclusion of the respondent from the analysis. Double entries will be checked using the options offered by Google Forms, as well as verifying the personal codes assigned to the participants.

#### 2.11.3. Quantitative Statistical Methods and Qualitative Data Analysis

A mixed method approach using quantitative and qualitative measures will be used. Quantitative measurements will evaluate potential effectiveness, including a prospective collection of outcome data assessing changes in children’s behaviour, parenting skills, and family adjustment in caregivers. In addition, qualitative interviews will facilitate an understanding of the caregiver’s experience in taking part in the intervention and the cultural acceptability.

We will follow CONSORT guidelines for reporting and quantitative analysis. Statistical analysis will be undertaken by a researcher with relevant expertise and an analysis of quantitative measures will be conducted upon completion of the final measures, three months post-intervention.

All data will be analysed according to the intention to treat. Ideally, all participants will be followed up with, as has previously been performed. In case of a loss of follow-up or not all questions being answered, a non-responder analysis will be performed and those who did not attend a follow-up will be compared to the remaining. Missing data will be excluded from the analyses or imputed using the serial mean, depending on the amount of missing data.

The normality of the data distribution will be assured through a visual inspection of the histograms, Q-Q plots, and box plots. Cronbach’s alpha coefficients will be calculated to assess internal consistency.

For demographic data, continuous variables will be presented as means and standard deviations (SD), with a 2-sample *t*-test for comparison; whereas, categorical data will be summarised as frequencies and proportions and compared using a chi-square test. In case of non-normality, the respective non-parametrical tests will be used.

To compare scores at the different time points, we will conduct a two-way multi-level mixed ANOVA (time: within-subjects factor and group: between-groups factor), considering the subject and the territory random effects. This overall analysis will be followed by the simple effect analysis, if the interaction is significant, and by post hoc tests for the main effect of time, if significant, using use a method based on the false discovery rate (e.g., Holm’s correction), proportional to the size of the difference between the means.

Regression analyses (SEM) will also be conducted between the parental dimensions and the filial dimensions to evaluate the effect of parental change on the behaviour and emotional state of the child.

Stratified analyses will be conducted by gender, as well as for participants with worse scores (below the 33rd or above the 66th percentile) at baseline. Statistical significance will be set at a *p*-value lower than 0.05.

Qualitative data will be analysed using thematic analysis (TA) [31], where dominant themes will be identified through a close examination of the data. First, open coding will be carried out and an initial coding schedule will be devised in order to define each emerging theme. The coding manual will then be revised throughout the coding of the remaining transcripts. The original codes will frequently be combined or divided into further codes, depending on the emergent findings. Themes will be continually compared with newly coded interview transcripts to ensure that they are readily applicable to the data by using the research team’s familiarity with the text and coding manual to frequently assess and reassess how the codes are being applied to the raw data. The coding manual will be discussed within the research team and amendments will be made if necessary. The team will then develop a revised code set that includes the new and combined codes. NVivo9 software will be used to facilitate the analysis. The analysis will cease when the research team judges that thematic saturation has been reached [31].

#### 2.11.4. Data Monitoring, Quality Assurance and Potential Harms

The data monitoring committee (DMC) will consist of the project managers in each of the sites in Italy, as well as the research assistants on site. In addition, three programme (support) and evaluation managers at UNODC headquarters (HQ) will contribute. The DMC will continuously report to the programme manager and the Chief of the Section at the UNODC. The DMC is independent from any sponsors and all members declare that they have no competing interests.

We do not plan to perform any thorough interim analyses; however, data completeness and quality controls will be performed as soon as the baseline measurements are completed for both groups. We do not foresee the necessity to terminate the trial before the anticipated date as no harm, adverse events, and other unintended effects of the intervention or trial conduct are to be expected.

#### 2.11.5. Auditing

As this is a fairly brief intervention and trial, we have not planned to conduct any auditing trials and, hence, have not defined the frequency and procedures.

### 2.12. Ethics and Dissemination

#### 2.12.1. Research Ethics Approval

All procedures performed in studies involving human participants were in accordance with the ethical standards of the institutional and/or national research committee and with the 1964 Helsinki Declaration and its later amendments or comparable ethical standards. This study has undergone an internal review performed within UNODC HQ. We designed and conducted our study based on the ethical standards developed by UNICEF and have taken all procedures into account. According to these standards, the submission of a trial such as ours is not mandatory for submission to a National Ethics Review Board or Institutional Review Board; however, this study should be reviewed by at least 3 experts. In our case, we collaborated with a number of stakeholders, apart from the experts from the UNODC, including the University of Parma and the University of Parma, where at least 3 independent researchers had reviewed this study thoroughly. Nevertheless, the protocol of this study, including all tools was submitted to the Research Ethics Board University of Parma, Protocol number 0238338; and received approval.

#### 2.12.2. Protocol Amendments

We do not plan to make any amendments to the protocol, such as changes to eligibility criteria, outcomes, or analyses. If it would, however, be necessary to change any of these, we would communicate any change immediately to all relevant parties, such as all investigators in the field, as well as HQ, REC/IRBs, trial participants, trial registries, and journals.

#### 2.12.3. Consent or Assent

Consent will be obtained through caregivers completing a consent form at the beginning of their appointment to complete the questionnaires at the first evaluation meeting. In addition to written information being provided in the form of the participant information sheets for the parents, participants will be provided with a verbal explanation of the evaluation method at the first meeting and again when they attend the first data collection session. If participants are only able to give verbal consent or assent, this will be audio-recorded.

Children and youth will complete assent forms, complementing the consent forms of their caregivers, in case they are under the age of 18 years as this is the legal age in Italy.

#### 2.12.4. Confidentiality

The chief investigator and the research team will preserve the confidentiality of participants in accordance with the Data Protection Act of 1998. All questionnaire data from participants will be either collected through Google Forms or conventionally on paper, with the process evaluated using digital recorders for the interviews. Identifying material will be removed as soon as possible from transcripts and notes. Each participant will be assigned a unique identification number and this will be used on all paper questionnaires and databases into which the data will be entered. Electronic copies of transcripts will be held separately from digital recordings of interviews and the file containing the participants’ names and corresponding numbers. All data collected as part of the trial will be treated as confidential and will only be viewed by members of the trial team; anonymised data will be used wherever possible.

Facilitators will be assigned a number that will be used on all reflection sheets and evaluations. Facilitator numbers will be stored using a password-protected file. The project manager and senior research team will be the only members of the research team holding a record of the individuals’ names corresponding to the facilitators’ numbers.

#### 2.12.5. Declaration of Interests

All authors and investigators for the overall trial and each study site confirm that they have no financial, or other, competing interests.

#### 2.12.6. Access to Data

After data cleaning and analysis, the anonymised datasets will be made available in a publicly available open database (i.e., Mendeley Data repository, OpenTrials, EudraCT, ICPSR, etc.).

#### 2.12.7. Ancillary and Post-Trial Care

Families in the control group (=waitlist group) will receive the intervention after the completion of all data collection points. Post-trial care will be service-as-usual. We do not expect anyone to suffer harm from the intervention; but, in the past, we have heard feedback from families who wish to have additional sessions of the programme, even after completion. We are currently working on add-on modules for the programme and hope to be able to offer these to families through the facilitators. In addition, a mobile phone application for families is currently being developed; however, roll-out is unlikely before the start of this trial.

#### 2.12.8. Dissemination Policy

Findings will be disseminated nationally to investigators, facilitators, and interested families straight after the completion of the analysis. Internationally, results will be disseminated to policymakers through presentations, in peer-reviewed journals to fellow researchers, through the UNODC website to the public, and through reports to donors.

Standard authorship eligibility criteria will be applied; we do not intend to use professional writers for upcoming publications of results but rather compose an author team based on the investigators/researchers involved in the trial. Through publication of this manuscript, we already grant public access to the full protocol and, if requested, we can share the anonymised participant-level dataset, once available, as well as the statistical codes.

## 3. Summary

Italy, particularly given the implication of two universities in this process, represents an ideal location for trialling Family UNited, given the existence of strong examples of the promotion of prevention based on science. Italy has been a country engaged with the UNODC on the development and dissemination of the UNODC WHO International Standards on Drug Use Prevention and has several health authorities and university centres that have been implicated in EU (European Union)-centred prevention initiatives, including the European Drug Addiction Prevention Trial that developed UNPLUGGED [32], the European Drug Prevention Quality Standards (EDPQS) Toolkits [33], the European Prevention Curriculum (EUPC) [34], and related training materials. Such a model of implementation would be significant to further supporting European Union efforts by introducing a tool with high scalability potential, supporting the EU Drug Strategy 2021–2025.

While Family UNited was initially developed in line with the UNODC WHO International Standards on Drug Use Prevention, making it primarily a drug-prevention programme, the etiological model addressed by these standards addresses a vulnerability that is common to many other social and health outcomes beyond drug use. Such outcomes include violence, mental health, and other risky behaviours. Given the easier proximity (timewise) to assess indicators of violence and mental health, these would be given priority. By measuring attitudes to violent behaviour and parental skills, we intend to measure changes on this domain, potentially leading to improvements on the targets set for the Sustainable Development Goal (SDG) indicators 16.1.3 (prevalence of all forms of violence).

Even though this trial is in a high-income country, such results complement the existing experience of the feasibility and efficacy of Family UNited previously piloted in LMIC, with impact-related measures encouraging the adoption of such approaches globally and beyond the EU borders. We would hope this trial will offer the opportunity to further strengthen the positive change of the culture of prevention by adding an open source and readily scalable tool to the registries of evidence in order to support more and more families globally and, more valuably, further support member states in reaching their sustainable development goals on the road to 2030.

## 4. Trial Status

After submitting this protocol for peer-reviewed publication, we received approval from the ethics committee review started the trial already. 

## 5. Standards of Reporting

This manuscript was prepared based on the “SPIRIT 2013 Statement” for reporting protocols of clinical trials [35].

## 6. Protocol Version

Version 1; 17 November 2022.

## Figures and Tables

**Figure 1 ijerph-20-06548-f001:**
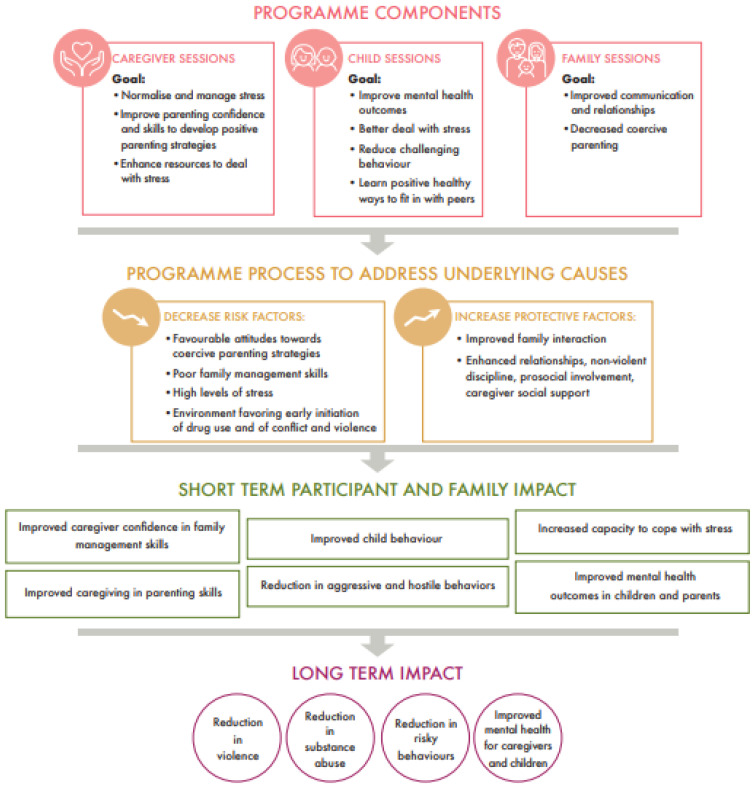
Family UNited logic model [19].

**Figure 2 ijerph-20-06548-f002:**
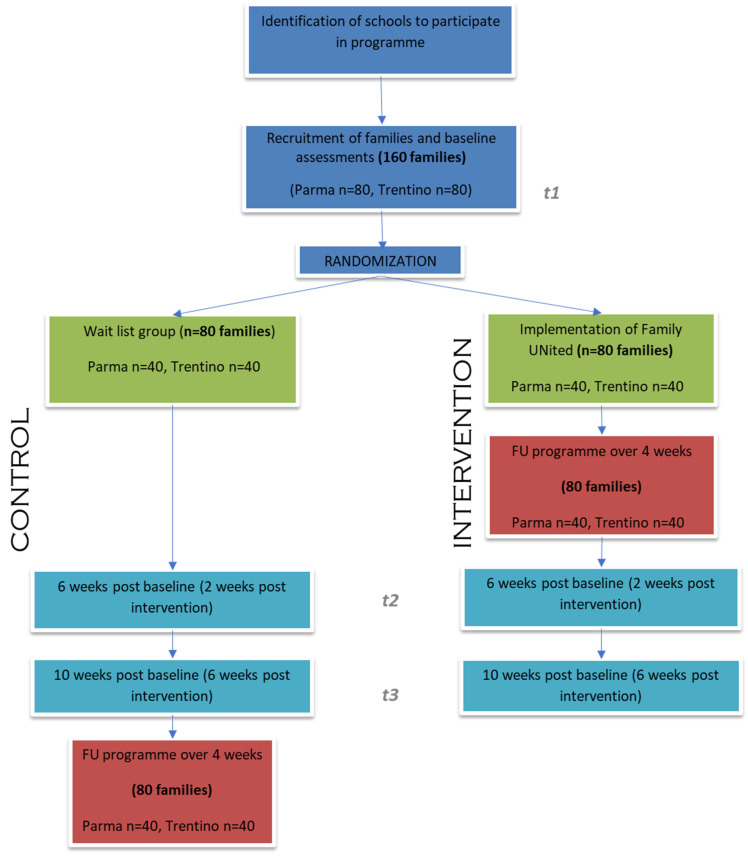
Family UNited RCT schema.

**Figure 3 ijerph-20-06548-f003:**
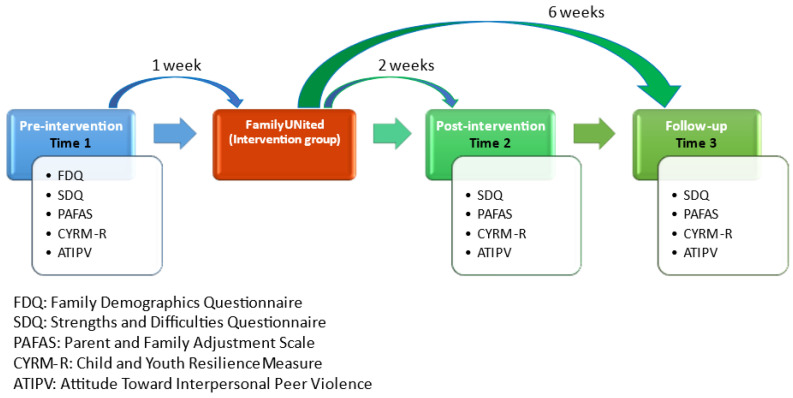
Data collection scheme.

**Table 1 ijerph-20-06548-t001:** Family UNited structure [19].

	Caregiver Session	Child Session	Family Session
**Week 1**	Understanding praising and encouraging children	Building positive qualities	Our family’s positive qualities
**Week 2**	Changing challenging behaviour	Handling stress	Learning about each other
**Week 3**	Responding to un-desirable behaviour	Skills to resist peer pressure I	Understanding peer pressure and family connections I
**Week 4**	Communicating and taking care of yourself	Skills to resist peer pressure II	Understanding peer pressure and family connections II

**Table 2 ijerph-20-06548-t002:** Dummy table of process evaluation components, data sources, and indicators.

Process Evaluation Component	Data Source	Indicator	Caregiver				Child				Family				Total
			Session 1	Session 2	Session 3	Session 4	Session 1	Session 2	Session 3	Session 4	Session 1	Session 2	Session 3	Session 4	
Dose delivered	Coordinator	Number of sessions delivered													
Fidelity	Observer	Percentage of activities reported as covered													
	Facilitator	Percentage of activities reported as fully/mostly covered as 3/4 (on a scale of 1 [not/hardly] to 4 [fully])													
	Observer	Percentage of programmes with ≥2 facilitators at every session													
	Observer	Percentage of programmes with ≥1 of the same facilitators at every session													
	Observer	Percentage of programmes with >4 and <15 families													
Dose received	Facilitator	Percentage of activities reporting interest of young people; and parents/carers as 3/4 (on a scale of 1 [low] to 4 [high])													
Reach	Observer	Percentage of families attending all 4 sessions													
Inputs	Observer	Percentage of sheets with a good or very good evaluation of the quality of childcare and travel arrangements													
	Observer	Percentage of sheets with a positive evaluation of (area of) refreshments													
	Observer	Percentage of sheets with a positive evaluation of room/materials/equipment													

**Table 3 ijerph-20-06548-t003:** Timeline of the pilot study and RCT.

Week	Activity
	PILOT STUDY
Week 1	Recruitment of 2 Research Assistants and training on measure completion
Week 2	Research Assistants recruit 10 families.
Week 2	Research Assistants conduct baseline measures and consent (Time 1) with all 10 families
Week 3–6	Families take part in weekly sessions of Family UNited for 4 weeks
Week 6	All families’ complete measures again (Time 2)
	RCT
Week 1–2	Research Assistants assess families for eligibility and inform them about the planned trial and the programme verbally and in writing
Week 3	Research Assistants recruit 160 families, obtain written assent/consent and conduct baseline measures (Time 1; pre-intervention) with all families After data collection has been completed, families are randomly allocated to intervention or control group
Week 4–7	Intervention group families take part in weekly sessions of Family UNited for 4 weeks Facilitators and observers complete process evaluation tools after each session
Week 9	ALL families complete measures again (Time 2; post-intervention). 10 caregivers from the intervention group are interviewed for acceptability
Week 13	ALL families complete measures again (Time 3; follow-up)
Week 14–17	Control families take part in weekly sessions of Family UNited for 4 weeks

**Table 4 ijerph-20-06548-t004:** Sample size estimates based on selected (statistically significant in paired *t*-test *) results from the pilot trial for Family UNited in Parma and Trentino, Italy, in spring 2022 [25].

	Before: Mean ± SD	After: Mean ± SD	Paired Difference in Means ± SD	Sample Size (Power)	
PAFAS				80%	90%
Positive encouragement (*n* = 14)	5.1 ± 2.3	2,1 ± 1.2	3.0 * ± 2.7	**10**	**12**
Parental adjustment (*n* = 14)	5.9 ± 3.1	4.7 ± 2.7	1.2 ± 2.7	**40**	**54**
Family relationships (*n* = 14)	3.8 ± 2.3	2.9 ± 1.7	0.9 * ± 1.2	**17**	**21**
SDQ					
Hyperactivity (*n* = 14)	2.8 ± 2.1	2.1 ± 1.8	0.7 * ± 1.1	**23**	**30**
Conduct problems (*n* = 14)	2.6 ± 1.8	1.3 ± 1.1	1.4 * ± 1.4	**12**	**15**
Total Difficulty Scale (*n* = 14)	10.9 ± 5.0	8.4 ± 4.4	2.6 * ± 3.7	**20**	**26**
CYRM-R					
Total CYRM (*n* = 12)	58.8 ± 19.0	60.8 ± 18.2	1.9 ± 6.9	**104**	**138**
Personal resilience (*n* = 12)	35.8 ± 11.8	37.4 ± 11.3	1.7 ± 3.9	**45**	**61**

## Data Availability

After data cleaning and analysis, the anonymised datasets will be made available in a publicly available open database (i.e., Mendeley Data repository, OpenTrials, EudraCT, ICPSR, etc.).

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
