# Peer review of "Assessing the Efficacy of a Brief Universal Family Skills Programme on Violence and Substance-Use Indicators in Youth in Trentino and Parma, Italy: Study Protocol for a Multi-Centre, Non-Blinded, Cluster-Randomised Controlled Trial (cRCT) of Family UNited"

_ijerph, 2023, doi:10.3390/ijerph20166548_

Round 1
Reviewer 1 Report
I congratulate the Authors of the idea and the initial results, which will be continued.
The abstract and keywords provide a good introduction to the topic of the article.
In the summary, the Authors explain that families experiencing stressful and difficult situations may not be able to cope with providing adequate childcare. This causes many negative social and health consequences during children's development.
Family Skills Training offers to strengthen family protective factors such as communication, trust, problem solving skills, therefore the Authors have developed a universal open source family skills package "Family UNIted" (FU) for families with children aged 8 to 15 living in low- and middle-income countries. The present authors' study aims to assess the effectiveness, fidelity and acceptability of FU in Trentino and Parma, Italy. This study also aims to assess improvements in parenting skills and family mental health after participating in FU compared with no intervention.
In the introduction, the authors explain that the Family Skills Packs are designed for reducing problems in family relationships and preventing future difficulties, and many of them are universal and are addressed to all families, regardless of the problems they experience.
Line 75, it would be advisable to support this justification with data from the scientific literature.
The Authors then explain the three overarching Family UNIted theories upon which they shaped program session elements. The logical model based on the Family UNIted approach - a universal preventive intervention to improve parenting skills - is presented by the authors in Figure 1.
The research procedure is detailed and correctly described. Similarly, the principles of ethical conduct and waiting for approval by the Ethics Board of the University of Parma, which is not necessary for this study, are described.
Since these are preliminary results and the research will continue, in my opinion, please consider not introducing the "conclusion" chapter, but changing it to "summary" (line 267).
Author Response
We thank the reviewer for his/her time and highly appreciate the summary and assessment. We have made the proposed revisions and added the literature, as suggested.
Reviewer 2 Report
Thank you for the opportunity to review this well-written paper. There are a few minor points which could be addressed before publication:
Intro:
- line 58: 'through' not 'thorough'
Method:
- are COVID physical distancing measures still relevant? in section 2.3
- it would be good to see a bit more about the qualitative data analysis and references for your chosen approach. Are you using a framework analysis method, thematic analysis etc? Will you use any software to facilitate analysis? This seems under-developed in comparison to the quantitative section.
- How will you integrate or compare the findings from the qualitative and quantitative data?
Author Response
We thank the reviewer for his/her time and highly appreciate the summary and assessment. We fully agree and have made the proposed revisions and added the additional information on the qualitative data analyses and integration.
Reviewer 3 Report
The manuscript is aimed at assessing the efficacy of a training program intended for families living in Trentino and Parma, Italy.
The manuscript, and especially the Method section, is well-articulated. However, its scientific soundness is compromised since the training is yet to be implemented. Therefore, there are no Results to be accessed by the scientific community. Nonetheless, the project is interesting and it might be interesting reading the results once carried out.
Below I report a few (minor) comments:
The title of paragraph 1.2 mentions substance use prevention but the paragraph does not mention the phenomenon.
Overall, the references are only 33, and I would suggest that the authors quote more literature to render the manuscript more scientifically sound.
And a few typos:
Antisocial can be written in full (i.e., not anti-social).
Line 91: initiated THE protect youth? Or initiated TO protect youth?
Line 119: development and NOT developmental.
The English is well-written, there are just a few typos, which I have reported in the comments above.
Author Response
We thank the reviewer for his/her time and highly appreciate the summary and assessment. We have made the proposed revisions and added more literature, as suggested. We have also corrected the typos, many thanks for highlighting them!